# UPR-Induced miR-616 Inhibits Human Breast Cancer Cell Growth and Migration by Targeting c-MYC

**DOI:** 10.3390/ijms241713034

**Published:** 2023-08-22

**Authors:** Vahid Arabkari, Afrin Sultana, David Barua, Mark Webber, Terry Smith, Ananya Gupta, Sanjeev Gupta

**Affiliations:** 1Discipline of Pathology, Cancer Progression and Treatment Research Group, Lambe Institute for Translational Research, School of Medicine, University of Galway, H91 TK33 Galway, Ireland; vahid.arabkari@gu.se (V.A.); afrin.sultana@universityofgalway.ie (A.S.); davidcu09@gmail.com (D.B.); mark.webber@universityofgalway.ie (M.W.); 2Krefting Research Centre, Department of Internal Medicine and Clinical Nutrition, Institute of Medicine, University of Gothenburg, 40530 Gothenburg, Sweden; 3Molecular Diagnostic Research Group, College of Science, University of Galway, H91 TK33 Galway, Ireland; terry.smith@universityofgalway.ie; 4Discipline of Physiology, School of Medicine, University of Galway, H91 TK33 Galway, Ireland; ananya.gupta@universityofgalway.ie

**Keywords:** unfolded protein response, ER stress, miR-616, c-MYC, breast cancer

## Abstract

C/EBP homologous protein (CHOP), also known as growth arrest and DNA damage-inducible protein 153 (GADD153), belongs to the CCAAT/enhancer-binding protein (C/EBP) family. CHOP expression is induced by unfolded protein response (UPR), and sustained CHOP activation acts as a pivotal trigger for ER stress-induced apoptosis. MicroRNA-616 is located within an intron of the CHOP gene. However, the regulation of miR-616 expression during UPR and its function in breast cancer is not clearly understood. Here we show that the expression of miR-616 and CHOP (host gene of miR-616) is downregulated in human breast cancer. Both miR-5p/-3p arms of miR-616 are expressed with levels of the 5p arm higher than the 3p arm. During conditions of ER stress, the expression of miR-616-5p and miR-616-3p arms was concordantly increased primarily through the PERK pathway. Our results show that ectopic expression of miR-616 significantly suppressed cell proliferation and colony formation, whereas knockout of miR-616 increased it. We found that miR-616 represses c-MYC expression via binding sites located in its protein coding region. Furthermore, we show that miR-616 exerted growth inhibitory effects on cells by suppressing c-MYC expression. Our results establish a new role for the CHOP locus by providing evidence that miR-616 can inhibit cell proliferation by targeting c-MYC. In summary, our results suggest a dual function for the CHOP locus, where CHOP protein and miR-616 can cooperate to inhibit cancer progression.

## 1. Introduction

The endoplasmic reticulum (ER) is a crucial cellular organelle that plays an important role in the folding and maturation of proteins that transit through the secretory pathway. Stimuli that compromise the folding capacity of the ER activate an evolutionarily conserved pathway known as unfolded protein response (UPR). Three major signaling nodes that characterize the UPR are inositol-requiring enzyme 1α (IRE1), protein kinase RNA-like endoplasmic reticulum kinase (PERK) and activating transcription factor 6 (ATF6) [1]. During the resting state ER-luminal domain of PERK, IRE1 and ATF6 interact with the ER chaperone glucose-regulated protein 78 (GRP78); however, upon accumulation of unfolded proteins, GRP78 dissociates from these molecules, leading to their activation [2]. The functional outputs of active IRE1 are tailored to the stimulus via the combination of the unconventional splicing of XBP1 mRNA and regulated IRE1-dependent decay of transcripts (RIDD) [3]. While splicing of XBP1 mRNA is a cytoprotective response to UPR, RIDD has many context-dependent outcomes [3]. ATF6 is a type-II transmembrane protein with an N-terminal cytoplasmic domain containing a DNA-binding motif and a C-terminal ER-luminal domain that binds GRP78 [4]. During UPR, ATF6 is first transported from the ER to the Golgi compartment, where it is activated by intramembranous proteolysis mediated by site-1 and site-2 proteases, causing the release of the transcriptionally active N-terminal cytosolic domain, p50ATF6 [5]. During UPR, active PERK phosphorylates its downstream targets, such as the eukaryotic translation initiation factor 2α (eIF2α) and nuclear factor erythroid 2-related factor 2 (NRF2) [6,7]. Phosphorylation of eIF2α leads to a general translational block and preferential translation of a subset of mRNAs, including activating transcription factor 4 (ATF4) [7].

Phosphorylation of NRF2 by PERK attenuates Keap1-mediated degradation of NRF2 and promotes the expression of antioxidant enzymes through the antioxidant response elements (ARE) [6]. The increased protein folding capacity and degradation of misfolded proteins induced by coordinated efforts of UPR attempt to restore protein homeostasis and increase cell survival. However, if UPR-induced mechanisms fail to alleviate ER stress, UPR activates apoptosis.

MicroRNAs (miRNAs) are a family of short (~20–23 nucleotide), endogenous, evolutionary conserved, single-stranded RNA molecules that regulate gene expression in a sequence-specific manner [8]. During the past few years, work from several groups has revealed that all three branches of the UPR regulate specific subsets of miRNAs [9,10,11,12]. The modes of regulation include induction/repression by UPR-regulated transcription factors, such as ATF6, XBP1, ATF4 and NRF2, as well as IRE1-mediated miRNA degradation. The outcome of UPR-dependent miRNA expression is the fine-tuning of the ER homeostasis to modulate cellular adaptation to stress and regulation of cell fate. The gene encoding miRNA-616 is located in the second intron of CHOP/DDIT3. This genomic co-location of miR-616 with CHOP/DDIT3 suggests a role for miR-616 in UPR. The expression of miR-616 has been shown to be upregulated in androgen-independent prostate cancer [13,14], gastric cancer [15], non-small cell lung cancer (NSCLC) [16], hepatocellular carcinoma (HCC) [17] and gliomas [18]. MiR-616 targets tissue factor pathway inhibitor (TFPI-2) in prostate cancer [13]; PTEN in HCC and gastric cancer [19]; SOX7 in glioma, and NSCLC [18]. However, miR-616-3p has been reported to reduce XIAP expression and potentiate apoptosis in HUVECs [20]. LINC01614 promotes head and neck squamous cell carcinoma progression via the PI3K/AKT signaling pathway, and miR-616-3p can inhibit cancer progression by downregulating LINC01614 expression [21].

Ye et al. reported that miR-616-3p inhibited cell growth and mammosphere formation of breast cancer cells by suppressing GLI1 [22], while Yuan reported that miR-616-3p promoted breast cancer cell migration and invasion by targeting TIMP2 and regulating MMP signaling [23]. Overall, these observations suggest that miR-616 exhibits both oncogenic and tumor suppressor functions in a context- and/or tissue-dependent manner. Additional studies are warranted to further investigate the role of miR-616 in human cancers.

In this study, we evaluated the role of miR-616 in breast cancer. We show that the expression of miR-616 and CHOP (host gene of miR-616) is reduced in human breast cancer. The expression level of miR-616-5p was higher than miR-616-3p in human cancer cells and tissues. During conditions of UPR, the expression of both miR-5p/-3p arms of miR-616 and CHOP was increased in a PERK-dependent manner. The ectopic expression of miR-616 significantly suppressed cell proliferation and migration, along with a reduction of c-MYC expression. The miR-616 downregulates c-MYC expression via binding sites located in its protein coding region. Finally, we show that restoring c-MYC expression in miR-616-expressing cells rescued the growth inhibitory effects of miR-616. Our results establish a new and unexpected role for the CHOP locus by providing evidence that miR-616 can inhibit cell proliferation by targeting c-MYC. In summary, our results suggest that the CHOP locus generates two gene products, where CHOP protein and miR-616 can act together to inhibit cancer progression.

## 2. Results

### 2.1. Prognostic Value of the miR-616 in Human Cancers

Several reports have described an oncogenic role of miR-616 in human cancers, but it is also reported to exhibit tumor suppressor activity in certain cancers. We assessed the prognostic value of miR-616 in different cancer types using a KM plotter. We found that expression of miR-616 was associated with longer overall survival (OS) in cervical squamous cell carcinoma, Head–neck squamous cell carcinoma, uterine corpus endometrial carcinoma and Thymoma (Appendix A). Surprisingly, the increased expression of miR-616 was associated with shorter OS in esophageal adenocarcinoma, sarcoma, renal clear cell carcinoma, lung squamous cell carcinoma and hepatocellular carcinoma (Appendix A). Next, we used TNMplot to determine the expression of CHOP and miR-616 in normal and tumor tissue samples of the breast. We found that expression of CHOP and miR-616 was reduced in tumor samples as compared to tumor-adjacent normal tissue (Figure 1A,B). Further, the expression of miR-616 was associated with poor OS in the luminal A subtype and better OS HER2 subtype of breast cancer (Figure 1C).

### 2.2. Expression of miR-616-5p Is Higher than miR-616-3p

Several mammalian miRNAs are located within the introns of protein-coding or non-coding genes and are referred to as intronic miRNAs [24]. The genes in which these miRNAs are embedded are called host genes. C/EBP homologous protein (CHOP) is encoded by the DNA damage-inducible transcript 3 (DDIT3) gene. miR-616, an intronic miRNA, is localized in the first intron of the DDIT3 gene (Figure 2A). Both the 5′ and 3′ arms of the precursor duplex can generate mature miRNAs and are referred to as miRNA-5p and -3p [24,25].

The miRBase [26] has discontinued the use of miRNA/miRNA* nomenclature and use of miRNA-5p and -3p nomenclature [27], based solely on 5′- or 3′-arm derivation of the mature miRNA is now recommended. The sequences of mature miR-616, accession numbers and the previous names of miR-616-5p and -3p as per miRNA/miRNA* nomenclature are shown in Figure 2B. Indeed, both 5p and 3p arms of miR-616 have been shown to be expressed in human cancers [15,28]. First, we evaluated the expression of miR-616-5p and miR-616-3p in various human cancer cell lines by qRT-PCR. We observed that the endogenous expression of miR-616-5p was higher than miR-616-3p in all the cell lines tested (Figure 2C). Next, we analyzed the expression of miR-616-5p and miR-616-3p in the TCGA dataset from different human cancers using OncomiR (http://www.oncomir.org, accessed on 15 August 2022), a web-based tool for miRNA expression analysis. Our analysis revealed a higher expression of miR-616-5p as compared to miR-616-3p in all the cancer types evaluated (Appendix A). Of note, the ratio of miR-616-5p and miR-616-3p expression varied considerably across the different tissue types. These results suggest that both 5p and 3p arms of miR-616 are expressed where miR-616-5p accumulates at higher levels than miR-616-3p, but their ratio is controlled in a tissue-specific manner.

### 2.3. Increased Expression of miR-616-5p and miR-616-3p during UPR

It was initially proposed that intronic miRNAs are derived from the same primary transcript as their host genes, thereby leading to the co-expression and co-regulation of miRNA and cognate host gene [24]. Analysis of the expression of 175 miRNAs and their host genes across 24 different human organs reported a significantly correlated expression profile [29].

However, recent evidence shows significant discordant expression between intronic miRNAs and host genes. Two likely explanations for this discordant expression are: (i) miRNAs have their own independent promoters and (ii) crosstalk between microprocessor cleavage and splicing [30,31]. Next, we evaluated the co-regulation of miR-616 and its host gene (CHOP) during conditions of UPR. MCF7, BT474 and HCT116 cells were either left untreated or treated with thapsigargin (TG) or Brefeldin A (BFA) for 24 h and expression of CHOP, miR-616-5p and miR-616-3p was determined. We found that expression of CHOP, miR-616-5p and miR-616-3p was upregulated in all three cell lines tested during conditions of ER stress (Figure 3A,B and Appendix A). These results suggest that miR-616 is co-transcribed along with the CHOP transcript.

Next, we investigated the role of ATF6, PERK and IRE1, three key mediators of the UPR, in the regulation of miR-616 expression. For this purpose, we used MCF7 control (MCF7 PLKO) and knockdown of UPR sensors (MCF7 XBP1-KD, MCF7 PERK-KD and MCF7 ATF6-KD) sub-clones of MCF7 [32]. We observed the reduction in the expression (basal and BFA-induced) of cognate target genes in the presence of the corresponding shRNA (Appendix A). We found that BFA-induced increase in the expression of CHOP, pri-miR-616, miR-616-5p and miR-616-3p was attenuated in PERK-knockdown sub-clones of MCF7 (Figure 4), whereas knockdown of ATF6 or XBP1 had no significant effect (Figure 4). Next, we generated the HCT116-control (HCT116 PKLO), HCT116 XBP1 knockdown (HCT116 XBP1-KD), HCT116 PERK knockdown (HCT116 PERK-KD) and HCT116-ATF6 knockdown (HCT116 ATF6-KD) sub-clones of HCT116 cells. We observed the reduction in the expression of cognate target genes due to the expression of the corresponding shRNA (Appendix A). We found that a BFA-mediated increase in the expression of pri-miR-616, miR-616-5p and miR-616-3p was attenuated in PERK-knockdown sub-clones of HCT116 cells (Appendix A), whereas knockdown of ATF6 or XBP1 had no effect (Appendix A). These results suggest that attenuated PERK signaling is required for the optimal induction of CHOP, pri-miR-616, miR-616-5p and miR-616-3p during UPR.

### 2.4. miR-616 Targets c-MYC and Reduces Cell Proliferation of Breast Cancer Cells

Next, we used gain-of-function (overexpression) and loss-of-function (CRISPR-Cas9 KO) approaches in MCF7 cells to study the role of miR-616 in breast cancer. To better mimic the in vivo scenario and express both miR-616-5p and -3p concurrently, we decided to use a plasmid-based system for overexpression. For this purpose, MCF7 cells were transduced with lentivirus expressing GFP along with miR-616 or Cas-9 along with gRNAs targeting miR-616. The expression of pri-miR-616, miR-616-5p and miR-616-3p was determined in miR-616-overexpressing (MCF7-miR-616-OE) and miR-616-knockout sub-clones of MCF7.

As expected, the levels of pri-miR-616, miR-616-5p and miR-616-3p were increased in the MCF7-miR-616-OE sub-clone (Figure 5A) and decreased in MCF7-miR-616 knockout sub-clone (Figure 5E). The MCF7-miR-616 knockout sub-clone displayed a hypomorphic phenotype with a partial reduction in expression of pri-miR-616, miR-616-5p and miR-616-3p. This could be due to the fact that the complete loss of miR-616 compromises the fitness and survival of MCF7 cells in tissue culture. The miR-616 knockout clone will be referred to as the miR-616 knockdown (miR-616-KD) sub-clone. The miR-616 overexpressing (MCF7-miR-616-OE) sub-clone showed significantly decreased cell growth (Figure 5B), while the knockdown (miR-616-KD) sub-clone showed increased cell growth (Figure 5F). This observation was further validated by colony formation assay. The miR-616 overexpressing (MCF7-miR-616-OE) sub-clone of MCF7 cells showed a decrease in the number and size of the colonies (Figure 5C,D), but the knockdown (miR-616-KD) sub-clone showed an increase in the number and size of the colonies (Figure 5G,H). To identify the genes regulated by miR-616, we performed transcriptomic analysis of control (MCF7-miR-CTRL) and miR-616-overexpressing (MCF7-miR-616-OE). We found that 276 genes were expressed at significantly lower levels, and 610 were expressed at significantly higher levels in miR-616 expressing MCF7 cells (Appendix A). A list of 5 genes was selected from the list of genes downregulated in miR-616 expressing MCF7 cells for further analysis by real-time RT-PCR. We found that expression of BCL2, c-MYC and VAV1 were downregulated in miR-616 expressing sub-clones of MCF7 and HCT116 cells (Appendix A).

On the basis of the fold change in expression and function, c-MYC was selected for further analysis. We observed that expression of c-MYC mRNA and protein was reduced in miR-616 overexpressing (MCF7-miR-616-OE) sub-clone of MCF7 cells (Figure 6A) and increased in miR-616 knockdown (MCF7-miR-616-KD) sub-clone (Figure 6B) when compared with respective control clones.

In agreement with the results in MCF7 cells, overexpression of miR-616 in HCT116 cells significantly decreased cell growth and migration that was accompanied by reduced expression of c-MYC (Appendix A). Since an inverse relationship between the levels of expression of a miRNA and its target is anticipated, we evaluated the expression of c-MYC during conditions of UPR. We found the expression of c-MYC was decreased in MCF7 (Figure 7A) and BT474 (Figure 7B) cells treated either with TG or BFA. Furthermore, expression of the c-MYC protein was decreased in MCF7 and BT474 cells upon treatment with BFA (Figure 7C). These results suggest a role for miR-616 in the regulation of c-MYC expression during conditions of UPR.

### 2.5. miR-616 Modulates the Expression of c-MYC via Site in the ORF

To determine the mechanism of c-MYC regulation by miR-616, we used several bioinformatics tools to search for the potential miR-616 binding sites in the reference sequence of c-MYC. TargetScan [33] (predicts targets of miRNAs by searching for the presence of conserved 8mer, 7mer, and 6mer sites that match the seed region of each miRNA) did not identify miR-616 binding sites in c-MYC. Considering the robust downregulation of c-MYC by miR-616, next, we used bioinformatics tools to search for miR-616 binding sites in 5′ UTR, ORF and 3′ UTR of c-MYC. Interestingly, RNAhybrid identified non-canonical miR-616-5p binding sites (at position 77, 2501 and 3076 bp) and miR-616-3p binding sites (at position 614, 917 and 1525 bp) of c-MYC reference Sequence (NM_002467.6) (Appendix A).

Furthermore, (miRWALK and RNAhybrid) picked an identical, non-canonical miR-616-3p binding site in the protein-coding region (364-1728 bp) of c-MYC at position 1525 bp (Figure 8A,B). To measure whether miR-616 down-regulates c-MYC via their coding region, we co-transfected 293T cells with an expression vector containing the coding sequence of c-MYC (without its 3′ UTR and 5′ UTR) and miR-616 plasmid. Cell lysates were collected after 24 h and analyzed by immunoblotting. Our results showed that the c-MYC protein level was strongly down-regulated by the co-expression of miR-616 (Figure 8C). Unlike miR-616, co-transfection of miR-4726 (c-MYC protein coding region lacks miR-4726 binding sites) did not decrease the expression of c-MYC protein level (Figure 8D). Further, co-transfection of miR-616 had no effect on the protein level of ERα (the protein-coding region of ERα has no predicted binding sites for miR-616) (Figure 8E). Taken together, these results suggest that miR-616 regulates the expression of c-MYC, most likely via the miR-616 response elements present in its protein coding region.

### 2.6. Expression of c-MYC Can Reverse the Growth Inhibitory Effects of miR-616

c-Myc is an oncogene frequently overexpressed in human tumors [34]. In light of our results, we reasoned that c-MYC could be the functionally relevant target that mediates the tumor-suppressive effects of miR-616. If suppression of c-MYC by miR-616 is indeed crucial for the tumor-suppressive effects of miR-616, overexpression of c-MYC should rescue the effect of miR-616 on cell growth. To test this, we transduced lentivirus that expresses c-MYC into miR-616 expressing MCF7 cells (Figure 9A). We found that the expression of c-MYC significantly reversed the growth inhibitory effects of miR-616 in MCF7 cells (Figure 9B). Further, we observed that the expression of c-MYC rescued the inhibitory effects of miR-616 on the number and size of the colonies in MCF7 cells (Figure 9C). Next, we determined whether c-MYC can rescue the effects of miR-616 on the migration of cells.

The bright field images of the wound healing revealed that expression of c-MYC restored the rate of wound healing in miR-616 expressing MCF7 cells that were comparable to control cells (Figure 9D,E). Thus, overexpression of c-MYC rescued the inhibitory effects of miR-616 on cell growth, colony formation and migration. The data, therefore, suggests an essential role for c-MYC as a mediator of the biological effects of miR-616 in breast cancer cells.

## 3. Discussion

MiRNAs regulate a wide range of cellular processes because of their ability to alter post-transcriptional gene expression [35]. We have evaluated the role of UPR-regulated miRNA-616 in cell growth, proliferation and migration. In this work, we have shown miR-616, located in the intron of CHOP, is upregulated during ER stress in a PERK-dependent fashion (Figure 4).

ER stress and UPR activation contribute to the initiation and progression of human diseases, such as obesity, neurodegenerative disorders, diabetes, cancer, and cardiovascular disease [2,36]. Our results suggest a mechanism for increased expression of miR-616 in pathophysiological conditions where the role of UPR is implicated. We observed concordant expression of the CHOP and the mature miR-616, which suggests their transcriptional co-regulation (Figure 3 and Figure 4).

Several studies have shown that miR-616 acts as a tumor-promoting miRNA in human cancers [15,28,37]. However, our results show that miR-616 expression attenuates the proliferation and migration of breast cancer and colorectal cancer cells (Figure 5, Appendix A). Furthermore, we observed that expression of miR-616 was reduced in human breast cancers as compared to tumor-adjacent normal tissue (Figure 1).

In agreement with our results, miR-616-3p has been shown to potentiate apoptosis in HUVECs [20], inhibit head and neck squamous cell carcinoma progression [21] and inhibit cell growth and mammosphere formation of breast cancer cells [22]. Any small RNA having a G-rich 6mer seed sequence (nt 2–7 of the guide strand) can reduce the viability of cells when associated with an RNA-induced silencing complex. G-rich seed sequence mediates toxicity by targeting C-rich seed matches in the 3′ UTR of genes critical for cell survival, referred to as Death Induced by Survival Gene Elimination (DISE) [27]. A systematic study to evaluate the DISE activity of all 4096 possible 6mer seed sequences in six cell lines (three human and three mouse) revealed the mechanism underlying this toxicity, and a web-based algorithm (https://6merdb.org) can predict the activity of an RNA with a known 6mer seed [38]. Analysis of seed sequences from both 5p and 3p strands of miR-616 revealed that both strands could exhibit opposing effects on viability in a cell-type-dependent manner (Appendix A).

Interestingly, several miRNAs belonging to miR-371-373 Cluster (hsa-miR-371a-5p, hsa-miR-371b-5p, hsa-miR-372-5p and hsa-miR-373-5p) that share seed sequence with miR-616-5p, have been shown to reduce the progression and metastasis of colon cancer [39] as well as induction of cell cycle arrest [40]. The divergent effect of miR-616 in human cancers can be reconciled by considering the fact that miR-616 has the capacity to target tens to hundreds of different mRNAs, some of which may have opposing oncogenic or tumor-suppressive functions. We propose that the oncogenic or tumor-suppressive effect of miR-616 is determined by the relative abundance of oncogenic/tumor suppressor transcripts that can be regulated by miR-616 in a given cellular context. Indeed, context-dependent oncogenic and tumor suppressor roles have been documented for several miRNAs, including the miR-17-92 cluster [41,42]. This cluster maps to human chromosome 13q31, a region amplified in Burkitt’s lymphoma, diffuse large B-cell lymphoma, follicular lymphoma, mantle cell lymphoma, and lung cancer [41,43]. The expression of miR-17-92 is reduced in prostate cancer, and restoration of its expression showed a therapeutic benefit [21]. The transgenic miR-17-92 expression in intestinal epithelial cells inhibited colon cancer progression by suppressing tumor angiogenesis [21]. Taken together, these observations indicate the presence of context-dependent pro- and anti-cancer effects of miR-616 in human cancers.

The expression of c-MYC was downregulated in miR-616 expressing MCF7 and HCT116 cells (Figure 6, Appendix A). Studies on target recognition by miRNAs have shown that sequence complementarity at the 5’ end of the miRNA, the so-called “seed region” at positions 2 to 7, is a main determinant for target recognition [33]. However, a perfect seed match of its own is not a good predictor for miRNA regulation, and a number of studies have shown that miRNA can regulate the expression of target genes via the sites with a G:U wobble and/or mismatch in the seed region [33]. Further, a study using a cross-linking and immunoprecipitation method to experimentally identify microRNA target sites in an unbiased manner has reported a significant number of non-canonical sites [44,45]. Hence, the presence of perfect seed complementarity is not essential for the regulation of target genes by miRNA. In addition, target sites for endogenous miRNAs have been reported in ORFs and 5′ UTRs, but they are less frequent than those in the 3′ UTR [46,47].

We found that c-MYC has non-canonical binding sites for miR-616 in the protein-coding region of the transcript (Figure 8, Appendix A). Several studies have reported the presence of miRNA-binding sites in the protein-coding sequence of the genes. Indeed, miRNAs have been shown to regulate embryonic stem cell differentiation, DNA methylation, regulation of apoptosis, aortic development, and tumor suppression via the functional miRNA-binding sites in the protein-coding sequence of the target genes [46,47,48].

There is significant crosstalk between UPR and c-MYC where the IRE1-XBP1 axis and c-MYC form a positive feedback loop. Spliced XBP1 directly upregulates c-MYC while c-MYC, in turn, induces ERN1 (encoding IRE1α) and XBP1 expression. Further c-Myc physically interacts with spliced XBP1s and potentiates its transcriptional activity. Our results show an inverse relationship between the expression of miR-616 and c-MYC during conditions of UPR.

We found the expression of c-MYC was decreased while that of miR-616 increased during conditions of UPR. Our results suggest that miR-616 upregulation can fine-tune c-MYC expression during UPR via the miR-616 binding sites located in its protein coding region. Several studies have reported the upregulation of c-MYC in different cancer cells and tumors, such as neuroblastomas, lung, and breast cancer [49]. Our results show that overexpression of c-MYC can reverse the effects of miR-616 on cell growth and migration in MCF7 cells (Figure 9). These observations suggest that the effects of miR-616 on cell growth and migration are primarily mediated by the downregulation of c-MYC.

PERK has been shown to act as a haploinsufficient tumor suppressor, where the nature of its function is determined by gene dose [50,51]. The transient pause in protein synthesis due to eIF2α phosphorylation by PERK is beneficial by reducing the secretory load in the ER.

Phosphorylation of NRF2 by PERK attenuates Keap1-mediated degradation of NRF2 and promotes the expression of antioxidant enzymes through the antioxidant response elements (ARE) [52,53]. PERK signaling can upregulate the CHOP/DDIT3 transcription factor, which inhibits expression of the gene encoding anti-apoptotic BCL-2 to hasten cell death, in addition to enhancing the expression of pro-apoptotic BCL-2 members, such as BIM [53]. Studies in both cellular and animal models with CHOP gene deficiency have shed light on the pro-apoptotic role of CHOP during cellular stress [53]. Further, ectopic expression of CHOP was reported to induce a G1 cell cycle arrest [54]. CHOP maintains the integrity of the human hematopoietic stem cell (HSC) pool by eliminating HSCs harboring oncogenic mutations and decreasing the risk of leukemia. CHOP induction triggers apoptosis of premalignant cells to prevent malignant progression in a mouse lung cancer model. Hepatocyte-specific CHOP ablation increased tumorigenesis in high-fat diet-induced steatohepatitis and Hepatocellular carcinoma [55,56].

CHOP has been shown to promote cancer progression when fused with FUS/TLS or EWS protein by genomic rearrangement [57,58]. The FUS-CHOP oncoprotein has been shown to induce metastasis in an in vivo model of sarcoma [57]. Accumulating data suggest that CHOP impinges upon several aspects of cancer, including initiation as well as the progression of tumors [53]. Our results show that miR-616 suppressed cell proliferation, colony formation and migration of cancer cells through suppressing c-MYC expression. Our results establish a new and unexpected role for the CHOP locus by providing evidence that miR-616 can inhibit cell proliferation by targeting c-MYC.

## 4. Materials and Methods

### 4.1. Cell Culture and Treatments

Human breast cancer cells (MCF7, SKBR3, MDA-MB231 and BT474) were purchased from ECACC. Colorectal cancer cells (HCT116) were a kind gift from Dr. Victor E. Velculescu, Johns Hopkins University, USA. HEK 293T cells were from Indiana University National Gene Vector Biorepository. The MCF7, SKBR3, MDA-MB231, BT474 and HEK 293T cells were maintained in Dulbecco’s modified eagle’s medium (DMEM) (Sigma, London, UK, Cat #D6429). HCT116 cells were maintained in McCoy’s 5A modified medium (Sigma, London, UK, Cat #M9309), and all mediums contained 10% heat-inactivated fetal bovine serum (FBS) and 100 U/mL penicillin and 100 μg/mL streptomycin (Sigma, London, UK, Cat #P0781) with 5% CO_2_ at 37 °C. To induce endoplasmic reticulum stress in cultured cells, two pharmacological compounds were used, including Thapsigargin (TG) and Brefeldin A (BFA), at the indicated concentrations for the indicated time points. Thapsigargin (Cat #1138) and BFA (Cat #1231) were purchased from Tocris Bioscience. All reagents were purchased from Sigma-Aldrich unless otherwise stated.

### 4.2. Plasmid Constructs

The PERK shRNA expressing lentiviral plasmid was a kind gift from Dr. Piyush Gupta, Massachusetts Institute of Technology, Boston, USA. The XBP1 shRNA expressing lentiviral plasmid has been described previously [59]. The ATF6 shRNA expressing lentiviral plasmid (TRCN0000017853, TRCN0000017855 and TRCN0000017857) were purchased from Dharmacon GE Healthcare Life Sciences. miExpressTM precursor miRNA expression clones, miR-CTRL and miR-616 (pEZX-MR03 vector) were sourced from GeneCopoeia, Rockville, MD, USA. The lentiviral plasmids pLV [CRISPR]-hCas9:T2A: Puro-U6 > mir-616 expressing hCas9 protein and miR-616 targeting gRNA were obtained from VectorBuilder Inc., Chicago, IL, USA. The sequence of two miR-616 targeting gRNAs are 5′-GGAAATAGGAAGTCATTGGA-3′ and 5′-GTGTCATGGAAGTCACTGAA-3′. PCDH-Flag-c-MYC was a gift from Hening Lin (Addgene plasmid #102626; http://n2t.net/addgene:102626; RRID: Addgene_102626).

### 4.3. Generation of Stable Cell Lines

Lentiviral plasmids were used to generate stable miR-616 overexpressing sub-clones of MCF7 and HCT116 cells, miR-616 knockout sub-clones of MCF7 and knockdown sub-clones of HCT116 and MCF7 cells for UPR genes (PERK, XBP1 and ATF6). Lentivirus was generated by transfecting lentiviral plasmids along with packaging plasmids in 293T cells using jetPEI transfection reagent (Polyplus transfection, VWR International Ltd., Dublin, Ireland) as described previously. Cells were transduced with the lentivirus, and selection was performed with 1 µg/mL puromycin for 7 days. The percentage of miR-616 transduced cells was evaluated based on GFP expression under a fluorescence microscope.

### 4.4. RNA Extraction, Reverse Transcription Reaction and Real-Time Quantitative PCR

Total RNA was isolated using Trizol (Life Technologies, Carlsbad, CA, USA) according to the manufacturer’s instructions. Reverse transcription (RT) was carried out with 2 μg RNA and random primers (Promega, Madison, WI, USA) using ImProm-II™ Reverse Transcription System (Promega). A real-time PCR method to determine the induction of UPR target genes has been described previously. Primers and probes were designed by Integrated DNA Technology (IDT), listed in Appendix A. TaqMan Universal Master Mix II (Life Technologies, #4440047) was used for the PCR reaction, and samples were run on StepOnePlus Real-Time PCR Systems (Applied Biosystems, Waltham, MA, USA).

RPLP0 and GAPDH (for mRNAs) and RNU6B (for miRNAs) were used as reference genes to determine the relative expression level of target genes between treated and control samples using the 2^−ΔΔCt^ method.

### 4.5. Colony Formation Assay

The control and miR-616 expressing cells (1000 cells per well) were plated in 6-well plates and were grown for 14 days. Then, cells were washed twice with PBS and fixed with 10% formaldehyde for 5 min and stained with 0.5% crystal violet for 10 min. The number of colonies that had more than 50 cells were counted in five random view fields under a microscope, and the average number of colonies was achieved. Colony size was determined using ImageJ software.

### 4.6. X-CELLigence Cell Proliferation Assay

X-CELLigence experiments were performed using the Real-Time Cell Analyzer (RTCA) Dual Plate (DP) instrument according to the manufacturer’s instructions (Agilent.com, accessed on 21 June 2021). Briefly, instrument-specific specially designed gold microelectrodes fused microtiter 16-well cell culture plate (0.2 cm^2^ well surface area; 250 µL volume per well) was used to monitor the real-time changes expressed as Cell index (CI). Cell Index is defined as (Rn − Rb)/15 where Rn is the cell–electrode impedance of the well with the cells, and Rb is the background impedance of the well with the medium alone. The background impedance was measured by adding 50 µL of medium to each well of the E-Plate before seeding the cells. After seeding (2500 cells/well), the E-plate was incubated at room temperature for 30 min and then transferred to the instrument. Cell proliferation was monitored every 15 min for 50 h.

### 4.7. Scratch Wound Healing Assay

The control and miR-616 expressing cells (3 × 10^5^ cells/well) were plated in 6-well plates. After 24 h of growth, when they reached 70–80% confluency, the cell monolayer was scratched with a sterile 0.2 mL pipette tip across the center of the well. Each well was then washed twice with medium to remove the detached cells and then replaced with fresh medium. The scratch areas were then imaged at different time periods, including 0 (after creating the scratch), 12, 24 and 48 h and the area of the scratch was quantified by ImageJ software.

### 4.8. Western Blot Analysis

Western blotting procedures have been described previously. The nitrocellulose membranes were blocked with the specific blocking solution for 2 h at room temperature. The nitrocellulose membranes were then treated with specific primary antibodies, including PERK (Cell Signalling, Danvers, MA, USA, Cat #3192), ATF6 (Abcam, Hong Kong, China, Cat #ab122897), Spliced XBP1 (Bio Legend, San Diego, CA, USA, Cat #619501), c-MYC (Santa Cruz Biotechnology Cat # sc-40, Dallas, TX, USA) and β-Actin (Sigma, London, UK, Cat #A-5060) at 4 °C overnight. After washing three times with PBS/0.05%Tween solution, the membranes were incubated with appropriate horseradish peroxidase-conjugated secondary antibody at room temperature for 2 h. The membranes were then washed twice with PBS/0.05%Tween and once with PBS, and finally, the signals were detected using Western Lightening chemiluminescent substrate (Perkin Elmer, Groningen, The Netherlands, Cat #NEL104001EA).

### 4.9. Statistical Analysis

The data were analyzed using the software package SPSS 21.0 for Windows, and a two-tailed unpaired *t*-test was performed to determine any statistically significant differences between independent groups. Results with a *p* < 0.05 were considered statistically significant. All experiments were performed in triplicates.

## Figures and Tables

**Figure 1 ijms-24-13034-f001:**
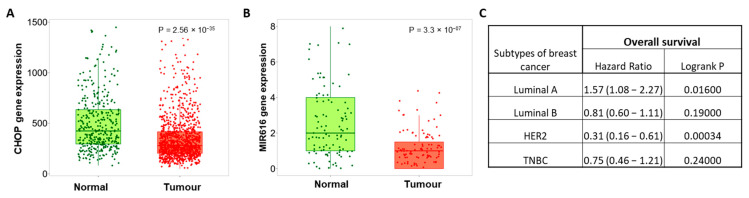
Prognostic value of miR-616 in subtypes of breast cancer. TNMplot (https://tnmplot.com/analysis/, accessed on 10 August 2022) was used to determine the expression of CHOP and miR-616 in human breast cancer. (**A**) Box plot for expression of CHOP in tumor (*n* = 1097) and normal (*n* = 403) tissues for human breast cancers is shown. (**B**) Box plot for expression of miR-616 in paired tumor (*n* = 112) and tumor-adjacent normal (*n* = 112) for human breast cancers is shown. (**C**) KM Plotter (https://kmplot.com/, accessed on 2 October 2022) was used to determine the association of miR-616 with overall survival (OS) in subtypes of breast cancer.

**Figure 2 ijms-24-13034-f002:**
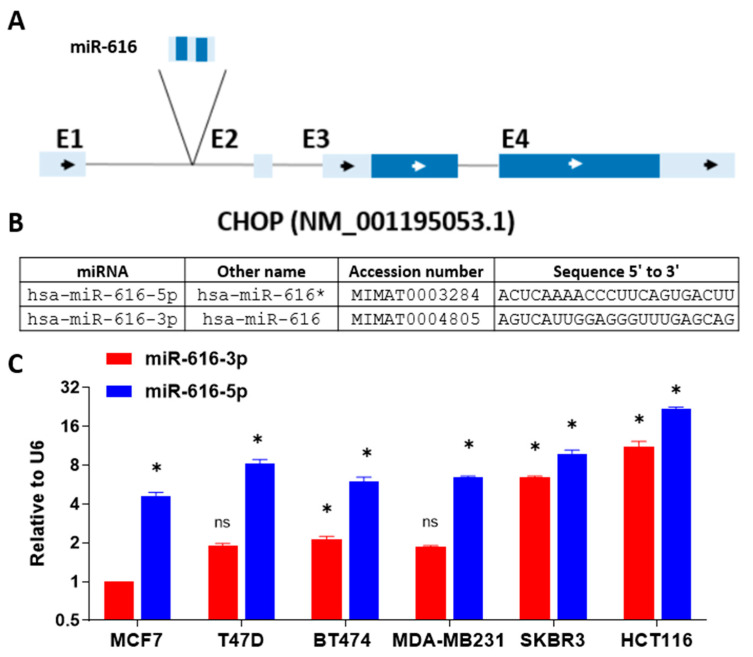
Basal expression of miR-616-5p and miR-616-3p in human cancer cell lines. (**A**) Schematic representation of the CHOP locus is shown. The exons are shown in blue, and the gray line between them represents the intron. The miR-616 is located in the first intron. The arrows show the direction of transcription of the gene. The protein coding region is shown in dark blue, and untranslated regions are shown in light blue. (**B**) The name, accession number and sequence of miR-616-5p and 3p are shown. (**C**) Total RNA from indicated cells was used to determine the expression of miR-616-5p and miR-616-3p by qRT-PCR and normalized against RNU6. Expression of miR-616-3p in MCF7 cells was arbitrarily set at 1. Error bars represent mean ± S.D. from three independent experiments performed in triplicate. * *p* < 0.05; ns, not significant. two-tailed unpaired *t*-test compared to miR-616-3p in MCF7 cells.

**Figure 3 ijms-24-13034-f003:**
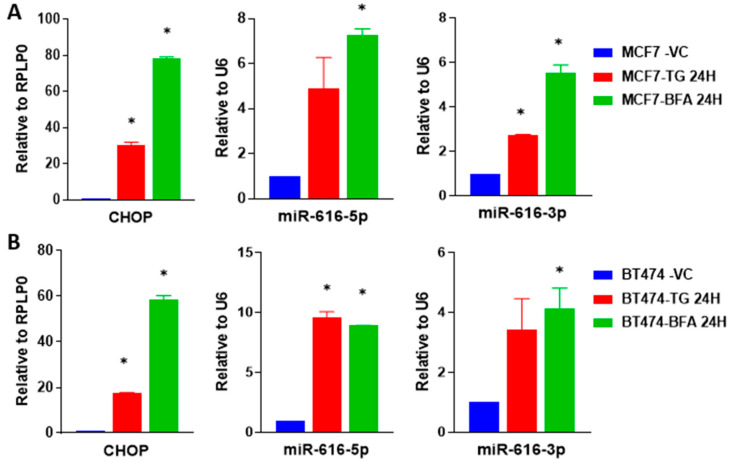
Upregulation of miR-616 (5p and 3p) and its host gene CHOP during UPR. (**A**) MCF7 cells and (**B**) BT474 cells were treated with (1 µM) TG or (0.5 μg/mL) BFA for 24 h. The expression of CHOP was quantified by RT-qPCR and normalized using RPLP0. The expression level of miR-616-5p and -3p was quantified by RT-qPCR, normalized using RNU6b. Error bars represent mean ± S.D. from three independent experiments performed in triplicate. * *p* < 0.05, two-tailed unpaired *t*-test as compared to vehicle control.

**Figure 4 ijms-24-13034-f004:**
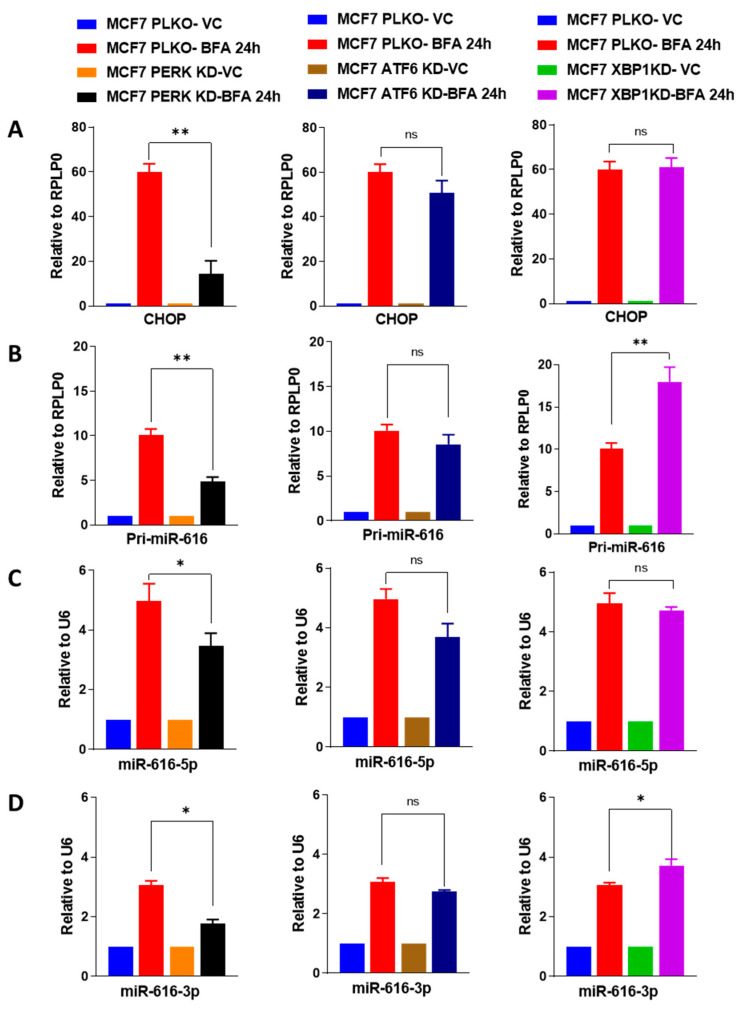
The upregulation of miR-616 during ER stress is dependent on the PERK pathway. MCF7 PLKO, MCF7 XBP1-KD, MCF7 PERK-KD and MCF7 ATF6-KD sub-clones were treated with (0.5 µg/mL) BFA for 24 h. The expression level of (**A**) CHOP, (**B**) pri-miR-616 was quantified by RT-qPCR, normalizing against RPLP0. (**C**,**D**) The expression level of miR-616-5p and -3p was quantified by RT-qPCR, normalized using RNU6. * *p* < 0.05, ** *p* < 0.001, two-tailed unpaired *t*-test. ns, not significant.

**Figure 5 ijms-24-13034-f005:**
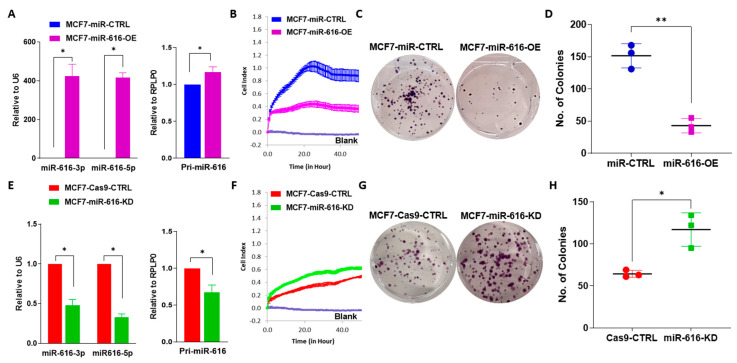
miR-616 reduces the proliferation of MCF7 cells. (**A**) The expression of miR-616-3p, miR-616-5p and pri-miR-616 is shown in MCF7-miR-CTRL and MCF7-miR-616-OE sub-clones of MCF7. (**B**) MCF7-miR-CTRL and MCF7-miR-616-OE cells were plated, and cell growth was determined by xCELLigence. (**C**,**D**) MCF7-miR-CTRL and MCF7-miR-616-OE cells were plated in a 6-well plate (1000 cells/well) and grown for 14 days. (**C**) Colonies stained with crystal violet are shown. (**D**) Quantification of the number of colonies for MCF7-miR-CTRL and MCF7-miR-616-OE cells is shown. (**E**) The expression of miR-616-3p, miR-616-5p and pri-miR-616 is shown in MCF7-Cas9-CTRL and MCF7-miR-616-KD sub-clones of MCF7. (**F**) MCF7-Cas9-CTRL and MCF7-miR-616-KD cells were plated, and cell growth was determined by x-CELLigence. (**G**,**H**) MCF7-Cas9-CTRL and MCF7-miR-616-KD cells were plated in a 6-well plate (1000 cells/well) and grown for 14 days. (**G**) Colonies stained with crystal violet are shown. (**H**) Quantification of number of colonies for MCF7-Cas9-CTRL and MCF7-miR-616-KD cells is shown. * *p* < 0.05; ** *p* < 0.001.

**Figure 6 ijms-24-13034-f006:**
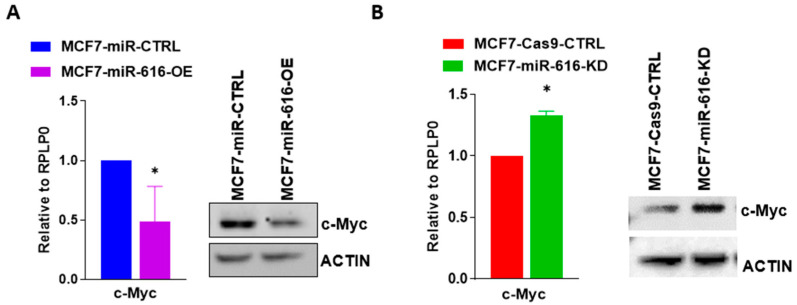
miR-616 regulates the expression of c-MYC. (**A**) The expression level of c-MYC mRNA and protein is shown in miR-616 overexpressing sub-clones of MCF7 cells. (**B**) The expression level of c-MYC mRNA and protein is shown in miR-616 knockdown sub-clones of MCF7 cells. Error bars represent mean ± S.D. from four independent experiments performed in triplicate. * *p* < 0.05, two-tailed unpaired *t*-test as compared to control.

**Figure 7 ijms-24-13034-f007:**
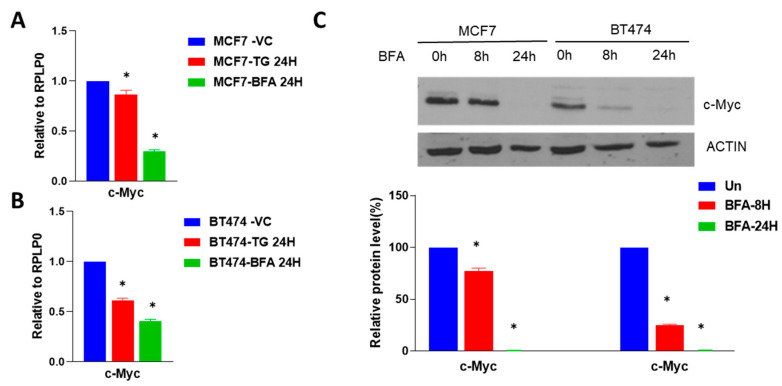
Downregulation of c-MYC during UPR in breast cancer cells. (**A**) MCF7 cells and (**B**) BT474 cells were treated with (1 µM) TG or (0.5 μg/mL) BFA for 24 h. Expression of c-MYC was quantified by RT-qPCR and normalized using RPLP0. Error bars represent mean ± S.D. from three independent experiments performed in triplicate. (**C**) MCF7 and BT474 cells were treated with (0.5 μg/mL) BFA for indicated time points. Western blotting of total proteins was performed using antibodies for c-MYC and actin. Representative immunoblot and quantification of c-MYC expression normalized to actin are shown (*n* = 3). * *p* < 0.05, two-tailed unpaired *t*-test as compared to the untreated control.

**Figure 8 ijms-24-13034-f008:**
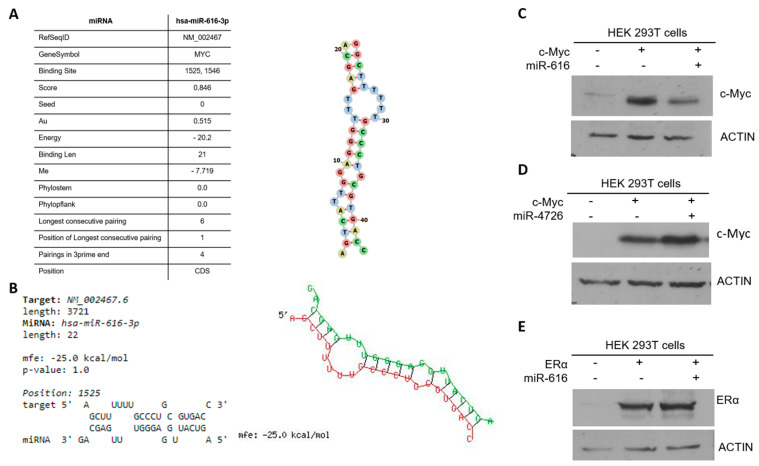
Identification of miR-616-3p-binding sites in the ORF of c-MYC. The binding site for miR-616-3p identified by (**A**) miRWALK and (**B**) RNAhybrid in the protein-coding region of the c-MYC reference sequence is shown. The secondary structure of miRNA-mRNA duplex is shown with miR-616-3p in green and c-MYC mRNA in red—mfe: minimum free energy. (**C**) 293T cells were transfected with a plasmid containing c-MYC ORF in the absence and presence of miR-616 plasmid. (**D**) 293T cells were transfected with a plasmid containing ERα ORF in the absence and presence of miR-616 plasmid. (**E**) 293T cells were transfected with a plasmid containing c-MYC ORF in the absence and presence of miR-4726 plasmid. Western blotting of total proteins was performed using the indicated antibodies. Twenty-four hours post-transfection, western blotting of total protein was performed using the indicated antibodies.

**Figure 9 ijms-24-13034-f009:**
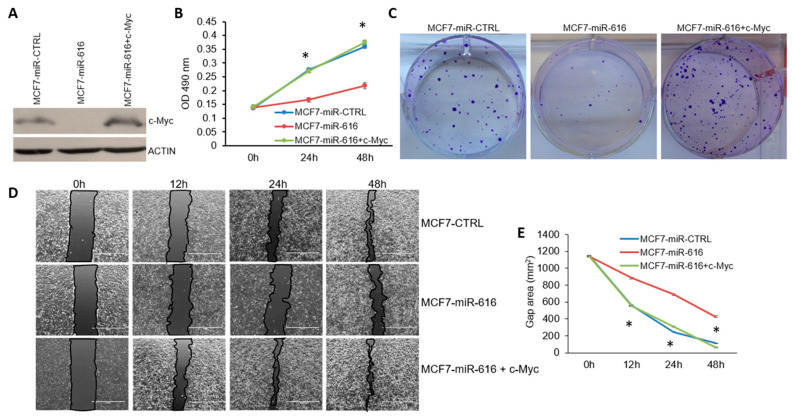
Restoration of c-MYC expression overcomes the inhibitory effect of miR-616 on cell proliferation. (**A**) The expression level of c-MYC protein in MCF7-miR-CTRL and MCF7-miR-616, and MCF7-miR-616 cells transduced with c-MYC lentivirus is shown. (**B**,**C**) Cell growth was determined by MTS assay and colony formation assay using MCF7-miR-CTRL and MCF7-miR-616, and MCF7-miR-616 cells expressing c-MYC. (**B**) Cell proliferation was assessed by MTS assay. Line graphs show the absorbance in cells at the indicated time points. Error bars represent mean ± S.D. from three independent experiments performed in triplicate. (**C**) Cells were plated in a 6-well plate (1000 cells/well) and grown for 14 days. Colonies stained with crystal violet are shown. (**D**,**E**) Cells were plated in a 6-well plate. The monolayer of cells was scratched with a micropipette tip (200 µL). Black lines indicate the wound borders at indicated time points post-scratching. Scale bar, 1000 µM. The line graph shows the scratch gap quantified using ImageJ software (v 1.8.0) at the indicated time points. Error bars represent mean ± S.D. from three independent experiments. * *p* < 0.05.

## Data Availability

Data is contained within the article or Appendix A.

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
