# Peer review of "UPR-Induced miR-616 Inhibits Human Breast Cancer Cell Growth and Migration by Targeting c-MYC"

_ijms, 2023, doi:10.3390/ijms241713034_

Round 1

Reviewer 1 Report

In manuscript IJMS-2497029, authors found miR-616 level was reduced in breast cancer and demonstrated miR-616 was induced by UPR through PERK-dependent pathway. Further study indicated miR-616 downregulated c-Myc through its ORF to suppress proliferation and migration in breast cancer cells. The manuscript was well written and layout is clear. However some assays only done in single cell line, authors need to have more data to support the conclusion. Below are specific comments:

Major Concerns:

(1) Throughout the manuscript MCF7, BT474 and HCT116 cell lines were used in most of the assays. However here the focus is breast cancer I suggest authors move HCT116 data to supplementary because it is a human colorectal carcinoma cell line. Also for each panel of the experiments authors need to use both MCF7 and BT474 cell lines to make conclusion solid and convincing.

(2) In Fig 1 and 2, authors concluded miR-616 could have prognostic value in breast cancer, authors should include MCF10 as control to compare normal vs. cancer in Fig 2C. Also authors could use the same cell line panel to show protein level of CHOP which could further validate the conclusion of Fig 1A and 1B.

(3) In Fig 4 or supplementary Fig 3&4, authors have to show PERK, ATF6 and XBP1 protein level to validate the knockdown in cells. Authors have to show the ER stress and UPR response upon BFA treatment.

(4) In Fig 6C, why was c-Myc level so different between Ca9 control and miR control, in theory these two controls should have relative same level of c-Myc. Also in Supplementary Fig 6D, why c-Myc had two bands but in other blots only single band?

(5) For Fig 9A, could authors replace with another blot because the current one showed odd pattern of the band.

Minor concerns:

(1) The font size in each figure should be uniformly same.

(2) The format of gene name should be all the same through the manuscript including figures, such as MDA-MB231 in the text and MDA-MB-231 in Fig2, MiR-616 in text and MIR616 in Fig2.

(3) Line 239, should that be xCelligence instead of x-xCelligence?

(4) Line 451, authors stated three compounds used but only listed TG and BFA.

Reviewer 2 Report

This manuscript by Arabkari et al., describing the CHOP, a member of the C/EBP family, is induced by unfolded protein response and plays a crucial role in ER stress-induced apoptosis. miR-616, located within the CHOP gene, is down-regulated in breast cancer and inhibits cell proliferation by targeting c-MYC. Manuscript is well written, however the data presented is not sufficient to support the conclusion of the paper.  1. There is no direct evidence showing miR-616 targets c-MYC. Authors need to do point-mutation in miR-616 and then check its effect on c-MYC expression by luciferase assay (10.1038/aps.2016.122). 2. There is no appreciable effect of miR-616 on MYC expression by qRT- (~10-30 % reduction figure 6A and B). Authors should explain the off-target effect of miRNA-CTRL on MYC expression (figure 6C, lane 3) 3. There are reports which show miR-166 activates c-MYC expression in lung cancer cells (10.3892/or.2017.5854). Authors should provide mechanisms for breast miR-166 mediated repression of c-MYC in breast cancer and discuss tissue specific discrepancy. 3. Figure 5E- Do authors create knockout for miR-616? Then why is mir-616 getting detected by qRTPCR? 4. Figure 5A, why did the authors plot CT value and not fold increased ?

Author Response

Reviewer 2 Comments and Suggestions for Authors

This manuscript by Arabkari et al., describing the CHOP, a member of the C/EBP family, is induced by unfolded protein response and plays a crucial role in ER stress-induced apoptosis. miR-616, located within the CHOP gene, is down-regulated in breast cancer and inhibits cell proliferation by targeting c-MYC. Manuscript is well written, however the data presented is not sufficient to support the conclusion of the paper.  

Author’s response: We thank the reviewer for the encouraging comments. Below we have addressed all the comments raised by the reviewer. 

  1. There is no direct evidence showing miR-616 targets c-MYC. Authors need to do point-mutation in miR-616 and then check its effect on c-MYC expression by luciferase assay (10.1038/aps.2016.122). 

Author’s response: The biological significance of a predicted miRNA-mRNA target pair can be validated by fulfilling four criteria (miRNA Targets: From Prediction Tools to Experimental Validation. Methods Protoc. 2021 Mar; 4(1): 1. Published online 2020 Dec 24. doi: 10.3390/mps4010001). First, the co-expression of miRNA and predicted target mRNA must be demonstrated. Second, a direct interaction between the miRNA of interest and a specific region within the target mRNA must be proved. Third, gain- and loss-of-function experiments must be performed to demonstrate how miRNAs regulate target protein expression. Fourth, it needs to verify whether the predicted changes in protein expression are associated with modified biological functions.

We agree with the reviewer that using 3’UTR reporter assays with wild-type and mutant miRNA-binding site is a method of choice to demonstrate the direct targeting of a given gene by a miRNA. However, this approach was not feasible because of the following reasons:

  1. We identified several non-canonical miRNA-binding sites for both miR-616-5p and miR-616-3p in the c-MYC reference Sequence (NM_002467.6) as shown in Fig 8 and SF7. Non-canonical miRNA-binding sites do not involve canonical pairing in the seed region (positions 2-7) of the mature miRNA and the rules for mutating the non-canonical miRNA-binding sites are not worked out.
  2. To better mimic the in vivo scenario and express both miR-616-5p and -3p concurrently we have used a plasmid-based system. It is not possible to introduce point mutations in miR-616 hairpin to disrupt the interaction with non-canonical miRNA-binding sites present in the c-MYC reference sequence without affecting the processing of hairpin to generate miR-616-5p and miR-616-3p.

Our results completely validate all the above-listed criteria except the second. We were not able to fully validate the second criteria due to the above-mentioned reasons but provide sufficient in-direct evidence of interaction between miR-616 and c-MYC reference sequence. We determined the effect of miR-4726 on the expression of c-MYC and the effect of miR-616 on the expression of ERα as controls to show the specific effect of miR-616 on the expression of c-MYC.

  1. There is no appreciable effect of miR-616 on MYC expression by qRT- (~10-30 % reduction figure 6A and B). Authors should explain the off-target effect of miRNA-CTRL on MYC expression (figure 6C, lane 3) 

Author’s response: We agree with the reviewer that ideally two controls (Cas-9 ctrl and miR-crtl) should have comparable levels of c-MYC expression. We do not have a proper explanation for this but most likely this is due to off-target effects control plasmids. Nonetheless, expression of c-MYC mRNA and protein was reduced miR-616 overexpressing (MCF7-miR-616-OE) sub-clone of MCF7 cells (Figure 6A and C) and increased in miR-616 knockout (MCF7-miR-616-KO) sub-clone (Figure 6B and C) when compared with respective control clones. We have added the following text in the revised manuscript.

Unexpectedly c-Myc expression level between MCF7-Cas9-CTRL control and MCF7-miR-CTRL was not the same. Most likely this is due to the off-target effects control plasmids. Nonetheless, expression of c-MYC mRNA and protein was reduced miR-616 overexpressing (MCF7-miR-616-OE) sub-clone of MCF7 cells (Figure 6A and C) and increased in miR-616 knockout (MCF7-miR-616-KO) sub-clone (Figure 6B and C) when compared with respective control clones.

  1. There are reports which show miR-166 activates c-MYC expression in lung cancer cells (10.3892/or.2017.5854). Authors should provide mechanisms for breast miR-166 mediated repression of c-MYC in breast cancer and discuss tissue specific discrepancy. 

Author’s response: We thank the reviewer for this suggestion. The reference (10.3892/or.2017.5854) shows that MicroRNA-616 promotes the growth and metastasis of non-small cell lung cancer by targeting SOX7. The effect of miR-616 on c-MYC expression in lung cancer cells is predominantly dependent on the regulation of SOX7 (a negative regulator of c-MYC) by miR-616.

The repressive activity of a miRNA varies according to target site accessibility and binding affinity. In addition, the repressive activity is also dependent on the relative cellular concentration of the miRNA and the total target pool (miRNA:target ratio) [Mol. Syst. Biol., 6 (2010), p. 363 & Genes Dev., 18 (2004), pp. 504–511 ]. Indeed mRNA repression can be modulated by perturbing endogenous miRNA:target ratios through overexpression of RNAs with multiple high-affinity miRNA binding sites, termed miRNA “sponges” [Nat. Methods, 4 (2007), pp. 721–726]. Further, miRNA target competition by sponges occurs in a threshold-like manner [Nat. Genet., 43 (2011), pp. 854–859] similar to other biological systems of molecular titration [Cell, 156 (2014), pp. 1312–1323 & Curr. Opin. Microbiol., 11 (2008), pp. 574–579]. The salient property of titration-mediated regimes in biology is nonlinear input-output responses that occur near the buffering molecule concentration. In the case of miRNAs, as the target pool surpasses the threshold set by the buffering miRNA concentration plus KD of the miRNA:target interaction, smaller changes in targets can result in larger changes in the concentration of free, i.e., unrepressed target [Nat. Genet., 43 (2011), pp. 854–859]. This relationship makes target affinity and cellular abundance important in determining responses to miRNA or target pool perturbations. For the sake of brevity, we have not included this in the revised manuscript.

The deletion or dysregulation of a miRNA is likely to have pleiotropic effects. The divergent effect of miR-616 in human cancers can be reconciled by considering the fact that any given miRNA has the capacity to target tens to hundreds of different mRNAs, some of which may have opposing oncogenic or tumor-suppressive functions. We speculate that the oncogenic or tumour-suppressive effect of miR-616 is determined by the relative abundance of oncogenes/tumour suppressor genes that can be regulated by miR-616 in a given cellular context.  Several other miRNAs have been shown to act as either an oncogene or a tumor suppressor in a context-dependent manner, such as miR-17-92, miR-155 and miR-125-b. MiR-155 has been shown to act as an oncogene in TNBC [Kong, W., He, L., Richards, E. et al. Upregulation of miRNA-155 promotes tumour angiogenesis by targeting VHL and is associated with poor prognosis and triple-negative breast cancer. Oncogene 33, 679–689 (2014). https://doi.org/10.1038/onc.2012.636] and as a tumor suppressor in ovarian cancers [MicroRNA-155 is a novel suppressor of ovarian cancer-initiating cells that targets CLDN1 FEBS Lett. 2013 May 2;587(9):1434-9. doi: 10.1016/j.febslet.2013.03.023. Epub 2013 Mar 20.]. MiR-125b acts as an oncogenic miRNA in hematologic malignancies, but as a tumor suppressor in many solid tumours [A.A. Svoronos, D.M. Engelman, F.J. Slack, OncomiR or Tumor Suppressor? the duplicity of microRNAs in cancer, Cancer Res 76 (2016) 3666–3670.]. MiR-17–92 exhibits context-dependent tumor suppressive or oncogenic effects in ER-positive or Triple negative breast cancer [Differential expression, function and prognostic value of miR-17-92 cluster in ER-positive and triple-negative breast cancer. Cancer Treat Res Commun . 2020;25:100224. doi: 10.1016/j.ctarc.2020.100224. Epub 2020 Oct 17.]. We have added following text in the revised manuscript.

The divergent effect of miR-616 in human cancers can be reconciled by considering the fact that miR-616 has the capacity to target tens to hundreds of different mRNAs, some of which may have opposing oncogenic or tumor-suppressive functions. We propose that the oncogenic or tumour-suppressive effect of miR-616 is determined by the relative abundance of oncogenic/tumour suppressor transcripts that can be regulated by miR-616 in a given cellular context.

  1. Figure 5E- Do authors create knockout for miR-616? Then why is mir-616 getting detected by qRTPCR? 

Author’s response: Our intention was to generate knockout of miR-616 in MCF7 cells but all our attempts were unsuccessful.  This was clearly mentioned in the manuscript   “The MCF7-miR-616-KO sub-clone displayed hypomorphic phenotype with a partial reduction in expression of pri-miR-616, miR-616-5p and miR-616-3p. Importantly partial reduction in miR-616 was sufficient to show biological effects.

We were not able to generate sub-clones of MCF7 with complete loss of miR-616. This could be due to the fact that the complete loss of miR-616 compromises the fitness and survival of MCF7 cells in tissue culture. More work is required to test this hypothesis which is beyond the scope of the current manuscript. We have added the following text in the revised manuscript.

This could be due to that fact that complete loss of miR-616 compromises the fitness and survival of MCF7 cells in tissue culture.

  1. Figure 5A, why did the authors plot CT value and not fold increased?

Author’s response: We thank the reviewer for this suggestion. We have made suggested changes to the revised manuscript.

Round 2

Reviewer 1 Report

The authors addressed all the points. 

Author Response

We thank the reviewer.

Reviewer 2 Report

1. As I asked in my previous report, there is no appreciable effect of miR-616 on MYC expression by qRT- (~10-30 % reduction figure 6A and B ). Authors need to explain this subtle change in MYC expression by miR-616. Normally   

2. Also at protein level, actin is also increased in the second lane (figure 6C) and if you normalize by loading control, cMYC expression is hardly increased, so the authors' claim for miR-616 regulate MYC expression have not substantial proof. Moreover, there are no direct evidence which add that miR-616 directly regulates MYC expression. 

3. According to authors claim, mir-4726 does not change MYC expression, but from the representative image figure 8D, MYC expression is increased by at-least ~50 %. 

4. Figure 5E, if authors could not get complete knockout of the gene of interest, strain should not be named as KO, it is confusing to the reader. 

Round 3

Reviewer 2 Report

1. Authors should provide the full blots for the new representative figures in 6B. 

2. Figure 8- Why did the authors suddenly opt for HEK293T cells instead of MCF7 or BT-474 cells to assess the impact of miR-4726 or miR-616? I highly recommend that the authors conduct these experiments using breast cancer cells such as MCF7 or BT-474. While HEK293T cells are known for their stability and significant protein expression due to SV40-T antigens, they are not ideal for investigating physiological functions.

Author Response

  1. Authors should provide the full blots for the new representative figures in 6B. 

Author’s response: We appreciate this concern and have added a full blot corresponding to figure 6B in the revised manuscript.

  1. Figure 8- Why did the authors suddenly opt for HEK293T cells instead of MCF7 or BT-474 cells to assess the impact of miR-4726 or miR-616? I highly recommend that the authors conduct these experiments using breast cancer cells such as MCF7 or BT-474. While HEK293T cells are known for their stability and significant protein expression due to SV40-T antigens, they are not ideal for investigating physiological functions.

Author’s response: We have used HEK-293T cells to show the regulation of c-Myc expression by miR-616 via presence of non-canonical miR-616 binding sites in the protein coding region of c-Myc. These analyses involved transient transfections with multiple plasmids to observe the anticipated effects and HEK-293T are the cells of choice for such analysis because of their ease of transient transfections, in addition to their stability and significant protein expression due to SV40-T antigens. We and others have successfully used HEK-293T cells for reporter assays in the past to show such regulatory mechanisms. We hope this reviewer will appreciate the technical challenge of the suggested experiments in hard to transfect cell lines. Due to time constraints for revision we were unable to repeat these experiments in MCF7 and BT474 cells. We sincerely believe that repeating these experiments in MCF7 and BT474 cells will not provide any new insights into the regulation of c-Myc by miR-616 and hope that this argument is acceptable.

Round 4

Reviewer 2 Report

NA

Author Response

na